# Novel Linalool-Silver nanoparticles: Synthesis, characterization, and dual approach evaluation via computational docking and antibacterial assays

Hina Manzoor[1], Mohammad A. Alfhili[2], Shakeel Waqqar[3], Malik Arslan Ali[1], Faiza Hassan[4], Samiullah Khan[5,6], Muhammad Umer Khan[1]*

1 Institute of Molecular Biology and Biotechnology, The University of Lahore, Lahore, Pakistan, 2 Department of Clinical Laboratory Sciences, College of Applied Medical Sciences, King Saud University, Riyadh, Saudi Arabia, 3 Department of Pathology and Laboratory Medicine, Auckland City Hospital, Auckland, New Zealand, 4 Chemistry Department, The University of Lahore, Lahore, Pakistan, 5 Faculty of Health and Life Sciences, INTI International University, Nilai, Negeri Sembilan, Malaysia, 6 Faculty of Pharmaceutical Sciences, University of Central Punjab, Lahore, Pakistan

* Muhammad.umer4@mlt.uol.edu.pk, umer.khan685@gmail.com

## Abstract

In recent years, scientists have developed new medical delivery techniques based on nanotechnology and have been actively creating nanoparticles combined with various extracts from natural plant products. This study aimed to conjugate linalool surface with silver nanoparticles, investigate its characteristics, and evaluate its effectiveness as a possible new therapeutic target against bacterial strains. A CMC-linalool solution was used to create linalool-based AgNPs (LN@AgNPs), which were then characterized by UV-Vis, FTIR, SEM, DLS, and zeta potential studies. The shape, hydrodynamic diameter, and negative zeta potential were among the advantageous properties of the resultant particles, which improved their performance and stability. Antibacterial potential was assessed using both *in silico* and *in vitro* methods. According to molecular docking, LN@AgNPs exhibit strong interactions via hydrogen bonding and potential metallic chelation with important bacterial protein residues (Cys, His, and Thr). According to the *in vitro* assays, LN@AgNPs exhibited inhibitory zones comparable to those of azithromycin and stronger antibacterial efficacy than free linalool against *Salmonella enterica*, *Bacillus subtilis*, and *Escherichia coli*. According to these results, LN@AgNPs may be a good option for creating potent antimicrobial agents. To improve its therapeutic application, further investigation of its mode of action and *in vivo* safety profile is necessary.

## 1. Introduction

Nanomaterials have garnered attention in various fields because of their distinctive physicochemical characteristics, including a high surface-area-to-mass ratio,

**Data availability statement:** All relevant data is available within the manuscript.

**Funding:** The author(s) received no specific funding for this work.

**Competing interests:** The authors have declared that no competing interests exist.

ultra-small dimensions, and substantial chemical action [1]. Carbon-based, inorganic, and organic nanoparticles are the three primary types of nanoparticles. Metal-based nanoparticles have a large surface area and small diameter. They are produced via destructive or constructive techniques using nanoscale metals. Almost all metals can be used to generate nanoparticles [2,3]. The use of silver nanoparticles, zinc oxide nanoparticles [4], copper nanoparticles [5], composite materials such as $Cu_2ZnSnS_4$ [6], and Pd-modified $MoS_2$ on Ti-based coatings [7] for the destruction of pathogenic microorganisms has been extensively studied in recent decades. One of the most frequently used metallic nanoparticles in contemporary antimicrobial applications is silver nanoparticles (AgNPs), which have broad bactericidal activity against both gram-positive and gram-negative bacteria in addition to good physicochemical characteristics [8]. When used in different situations, cationic silver, a common bactericide, has deleterious effects on microscopic biota [9]. AgNPs also show great potential for antibacterial applications, because they can inhibit the development of biofilms and eliminate a large number of drug-resistant strains [10]. Recent research has demonstrated that numerous molecules found in various bacteria, such as *Salmonella typhi, Escherichia coli, Staphylococcus aureus, Klebsiella pneumoniae, Citrobacter koseri, Bacillus cereus, Pseudomonas aeruginosa, and Vibrio parahaemolyticus*, bind to AgNPs and can be inhibited through a variety of mechanisms, such as AgNPs interacting inside cells, generating free radicals, and modifying transduction pathways [11].

In recent years, researchers have developed a novel drug delivery method based on nanotechnology, and have been involved in the creation of nanoparticles combined with various extracts from natural plant products. Various methods and instruments from other fields, including biology, chemistry, and medical research, have been employed in nanotechnology [12]. The underlying principle of nanoscale research may be the structural attribute of a molecule that is often unavailable in bulk solids and individual molecules. This sector has the potential to significantly increase medication bioavailability and in turn minimize or reduce the toxicity associated with high doses, which are typically required for the best possible outcome [13].

To understand the impact of naturally occurring monoterpene components in essential oils (EOs) on the human body, both in isolation and in amounts linked to currently prescribed drugs, extensive studies are being conducted on these elements. Linalool, a monoterpenoid, is commonly found in many herbs and is used to flavor black tea [14]. This material promotes the preservation of cosmetic formulations and exerts anti-inflammatory effects on mild skin lesions by providing a pleasant smell. Many essential oils (EOs), including *lavender*, *coriander*, and *basil*, contain linalool as the main active ingredient, which has potent antibacterial properties [15]. Another way to make people more susceptible to antibiotics is to use linalool as an adjuvant antimicrobial agent. Additionally, it has been shown to have antimicrobial properties [16,17]. Previous studies have shown that it is highly significant because of its wide variety of biological activities, including antioxidant, anticancer, anti-inflammatory, and antibacterial properties [15]. The mode of action of linalool against microorganisms is still under investigation.



The purpose of this study was to conjugate the linalool surface with AgNPs to examine its characteristics and investigate its potential as a cutting-edge therapeutic approach against bacterial strains.

## 2. Material and methodology

### 2.1. Material

Carboxy methylcellulose (CMC), Linalool with 97% purity, Silver nitrate (AgNO$_3$) and Sodium hydroxide (NaOH) were acquired from Sigma-Aldrich (Merck Group, Germany). All the reagents were analytically pure and did not require additional purification.

### 2.2. Synthesis of Linalool-Silver nanoconjuagte (LN@AgNPs)

The linalool-silver nanoconjugates were synthesized as described by Moustafa et al. [18] with some modifications. Deionized water (300 mL) and CMC (7 mg) were added to a 500 mL flask. The mixture was mechanically agitated for 15 min after 3 mL of linalool was added, and swirled at 70 °C to produce a homogeneous viscous solution. AgNO$_3$, a silver precursor, was added to deionized water and stirred for five minutes to produce 200 mL of a 0.05M solution. This solution was added to the first solution. After progressively adding NaOH to obtain a pH toward the base, the mixture was agitated with a magnetic stirrer for 15 min at 100 rpm. The mixture turned brown instead of colorless, indicating production of LN@AgNPs. To enable further characterization, a portion of the solution was centrifuged for 20 min at 5000 rpm and then dried for 48 h at room temperature. The remaining portion was used for microbiological examination.

### 2.3. Characterization of Linalool-based Silver nanoconjuagte (LN@AgNPs)

**2.3.1. Ultraviolet-visible absorption spectroscopy.** The absorbance of the samples was assessed using a Shimadzu UV-Vis spectrometer (UV-1800, Shimadzu, Japan). The LN@AgNPs UV-VIS spectra were recorded in the visible and ultraviolet spectral regions (200–800 nm). Distilled water was used as a reference for baseline adjustment. Spectral analysis confirmed the formation of LN@AgNPs [19].

**2.3.2. Particle size (PS) and zeta potential (ZP) measurements.** Important factors for stability investigations include the electrostatic potential and average PS of LN@AgNPs, which have been characterized [20]. After dilution with dH2O, PS and ZP of synthesized conjugated silvernanoparticles were assessed at 25 °C using an Anton Paar Zetasizer Nano ZS (Litesizer 500). The experiments were independently conducted thrice to reduce the possibility of mistakes [21,22].

**2.3.3. Fourier transform infrared spectroscopy (FT-IR) analysis.** Shimadzu FTIR, was used to acquire the IR spectra for chemical bonding determination. FTIR analysis was used to determine the functional groups responsible for the reduction and stability of the biosynthesized AgNPs. The study was conducted at wavenumbers of 400–4000 cm$^{-1}$ [23].

**2.3.4. Scanning electron microscopy (SEM).** An FEI Nova NanoSEM 450 (USA) microscope was used for morphological investigations. SEM was used to verify the crystalline state, shape, and size of LN@AgNps [24].

### 2.4. Evaluation of antibacterial activity of Ag@LN NPs (*In silico* and *In vitro* studies)

**2.4.1. Examine organisms.** To compare the antibacterial activity of LN@AgNPs with that of individual linalool, minimum inhibitory concentration (MIC) testing was performed on two gram-negative bacterial strains, *Salmonella enterica* and *Escherichia coli*, with ATCC numbers 14028 and 25922, respectively, and one gram-positive bacterial strain, *Bacillus subtilis* ATCC6633.

**2.4.2. Molecular docking analysis.** Using molecular docking analysis, the interactions between the synthesized LN@AgNPs and individual linalool with selected proteins from *Escherichia coli*, *Salmonella enterica*, and *Bacillus subtilis* were examined. The structures of these proteins were determined using 3D atomic coordinates from the Protein Data Bank (PDB IDs: 6IO4, 3FHU, and 2VAM respectively). Protein structures were prepared for docking using AutoDock Vina 1.5.7.

This involves removing water and ligands and then adding polar hydrogen and charges (Kollman and Gasteiger). The structure of LN@AgNPs was built, and a Chem3D Gaussian interface was used to minimize its energy. Using Autodock Vina 1.5.7 [25], the constructed conjugated structure LN@AgNPs and individual linalool were docked into the target protein (6IO4, 3FHU, and 2VAM) active sites. The ligand interaction tool Discovery Studio 2021 [26] was used to view the interaction diagram of the ligand protein active-site residues. [27].

**2.4.3. Minimum inhibitory concentration (*In vitro*) determination.** A microplate bioassay was used to determine the MIC of LN@AgNPs in relation to individual linalool. Briefly, bacterial suspensions were prepared using a 0.5 McFarland standard. In a 96-well plate, 100 μL of nutrient broth was added to each well, followed by the bacterial suspension, except in the negative control wells. Then, 50 μL aliquots of linalool and LN@AgNPs were added to each well. In this experiment, a range of sample concentrations was employed in the wells from 50 to 6.25 μL/mL. The wells were incubated for the entire day at 37 °C. A PR 4100 microplate reader was used to measure the optical density (OD) at 560 nm to verify the antibacterial activity. The minimum inhibitory concentration (MIC) hinders the development of microorganisms. Five runs of each experiment were conducted and the results were consistent [28].

**2.4.4. Agar-well diffusion assay.** Linalool and LN@AgNPs were tested for antibacterial activity using an agar-well diffusion assay. In this study, we assessed the inhibition zones of linalool and LN@Ag NPs. Millimeters (mm) were used to measure the inhibitory zones. To inoculate Mueller-Hinton agar plates, three strains of *Salmonella enterica*, *Escherichia coli*, and *Bacillus subtilis* were introduced. A cork borer was used to drill 6 mm diameter wells into which 10 μL of the samples (Linalool and LN@AgNPs) were added. Azithromycin (AZM) was used as the positive control, whereas distilled water served as the negative control. The plates were incubated for a full day at 37 °C. Growth inhibition zone widths were assessed following incubation [29].

**2.4.5. Disc diffusion susceptibility assay.** The prepared medium was placed in sterile Petri dishes, aseptically moved, and allowed to set. The following hour, bacteria were cultured on uncontaminated plates. Bauer et al. (1966) described the disc diffusion assay method, which was used to assess the antibacterial activity of materials against a particular group of bacterial strains [30]. Microbial inoculums (standardized inoculums 1−2 × 107 CFUml-1 0.5 McFarland Standard) were seeded onto Mueller-Hinton agar plates, and the plates were allowed to develop for 18–24 h. Whatman No. 1 filter paper discs (6 mm in diameter) were positioned on top of the media in petri dishes using sterile forceps. The discs were treated with linalool and LN@AgNPs. Water (10 μL/disc) was used as a negative control, and AZM was applied to the discs as a positive control. The inoculated plates were incubated at 37 °C for 18–24 h. The following day, the zones of inhibition around the discs on each plate were measured in millimeters [31].

**2.4.6. Data analysis.** OriginPro 2024b (OriginLab Corporation, Northampton, MA, USA) was used to statistically assess the values after each measurement was performed in triplicate [32].

## 3. Results and discussion

The methods outlined in the methodology section were used to synthesize LN@AgNPs [18], with modifications. Fig 1 illustrates this process. To create a silver conjugate based on essential oil (linalool), silver nanoparticles (Ag NPs) were incorporated into the CMC-linalool biomaterial solution following the steps described in the methodology section. The hydroxyl groups (-OH) in linalool attract $Ag^+$ ions, which initiate their conversion into silver nanoparticles. Linalool functions as a capping and reducing agent to regulate the size of the nanoparticles, and CMC serves as a stabilizing agent. NaOH was then gradually added to bring the pH to a basic level. Hydroxyl ions ($OH^-$) from NaOH neutralize the solution and encourage the reduction of $Ag^+$ to metallic Ag (Ag°) nanoparticles. LN@AgNPs or linalool conjugated silver nanoparticles were formed when the color shifted from transparent to colorless to brown (Fig 1). The ability of phytochemicals to function as both capping and reducing agents is also supported by earlier research [33]. Functional groups included in phytochemicals, particularly hydroxyl (–OH) groups, are primarily responsible for mediating this reduction, as reported by Kaur et al. [34]. The nanoparticles performed better at higher pH values despite the fact that temperature variations had no effect

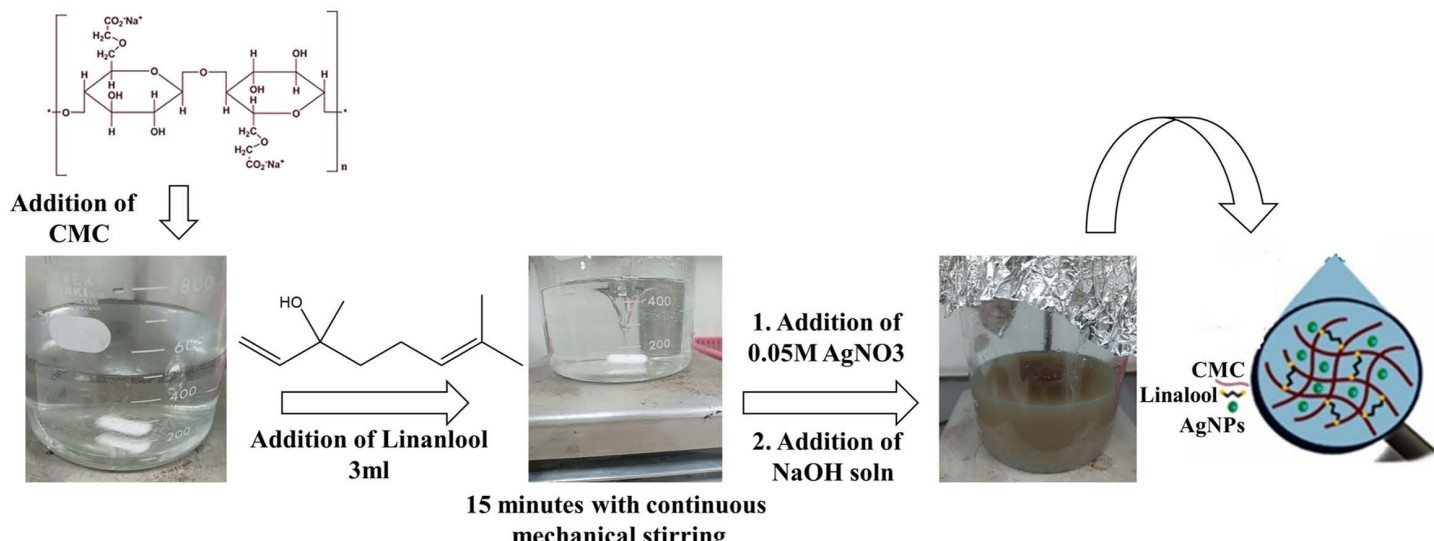

**Fig 1. Preparation Steps for LN@AgNPs (change of color from transparent to brown showed that nanoconjugation occurred).**

on the production process [35]. Therefore, the synthesis of AgNPs in the present study was carried out in an alkaline pH setting. For additional characterization, the solution was centrifuged at 5000 rpm and dried at 60℃ for 48 h.

### 3.1. Characterization of synthesized LN@AgNPs

**3.1.1. Ultraviolet-visible spectrometry.** The production of nanoparticles was verified and the relationship between the average diameter of the particles and the peak observed at a particular wavelength was evaluated using ultraviolet-visible spectroscopy. The dispersion of the nanoparticles was verified using the widths of the Surface Plasmon Resonance (SPR) bands. The size and shape of the NPs are indicated by the peaks in the SPR band. The dispersity and free electron density of the NPs were determined based on their bandwidths. A vast absorption peak appeared at λmax = 433 nm (Fig 2).

Our results agree with those of previous studies. According to Solomon et al., particles with an average diameter of 35–80 nm can be distinguished by wavelength peaks between 415 and 440 nm [36]. Cip- and Lev-conjugated AgNPs displayed peaks at 420 and 412 nm, respectively, while another study found that the maximum absorbance of turmeric leaf extract was 433 nm and that of moringa leaf extract was 445 nm [37]. The reduction of silver ions into silver nanoparticles causes surface plasmon resonance in the metal nanoparticles, which results in a color shift in the biologically produced nanoparticle solution [38].

**3.1.2. Particle Size (PS) and Zeta Potential (ZP) Measurements.** Dynamic light scattering was used to ascertain the hydrodynamic diameter of the AgNPs. The combined size of the metallic silver core and the biochemical coating of AgNPs, which are made up of functional groups first discovered in plant secondary metabolites, have been demonstrated [39]. The AgNPs were found to have a size distribution (by intensity) of approximately 10–1000 nm in Brownian motion (Fig 3a). A histogram was constructed to determine the size distribution of AgNPs (Fig 3b). The average diameter was determined using a Gaussian model and a nonlinear curve-fitting technique. 89.21 nm ± 0.59 nm is the average particle diameter for LN@AgNPs, based on the fitted curve. These results are similar to the UV-Vis findings. AgNPs with an orbicular shape and a particle size of less than 100 nm emerged as a yellow-to-brown colloidal liquid with an absorbance peak at 400 nm. This is consistent with the results of previous studies [40–42].

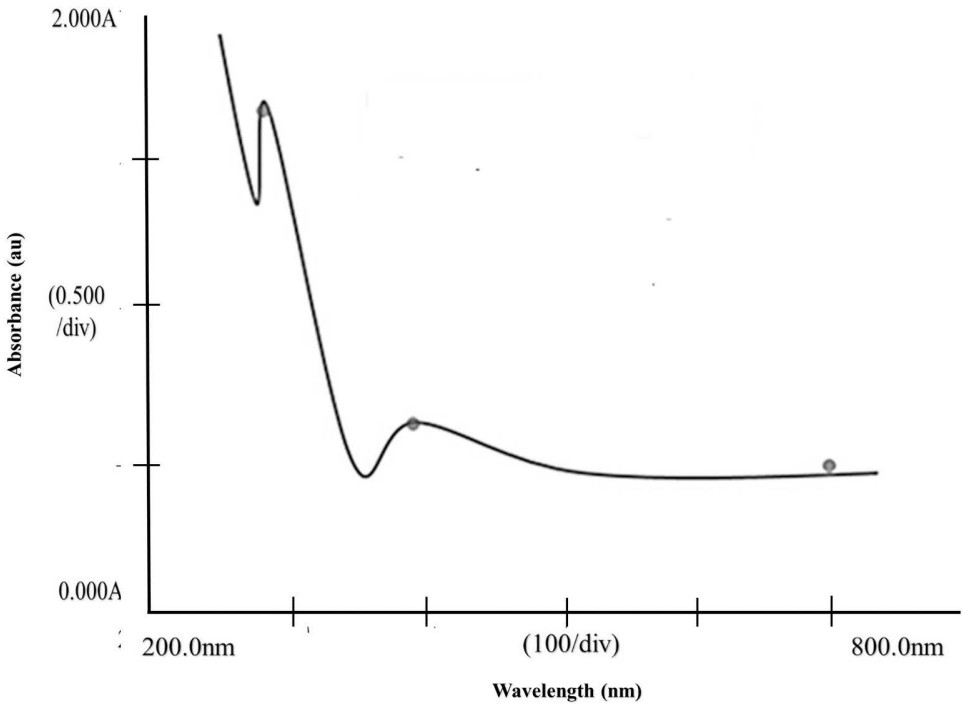

**Fig 2. Ultraviolet-visible spectroscopy: SPR bands spectrum of LN@AgNPs.**

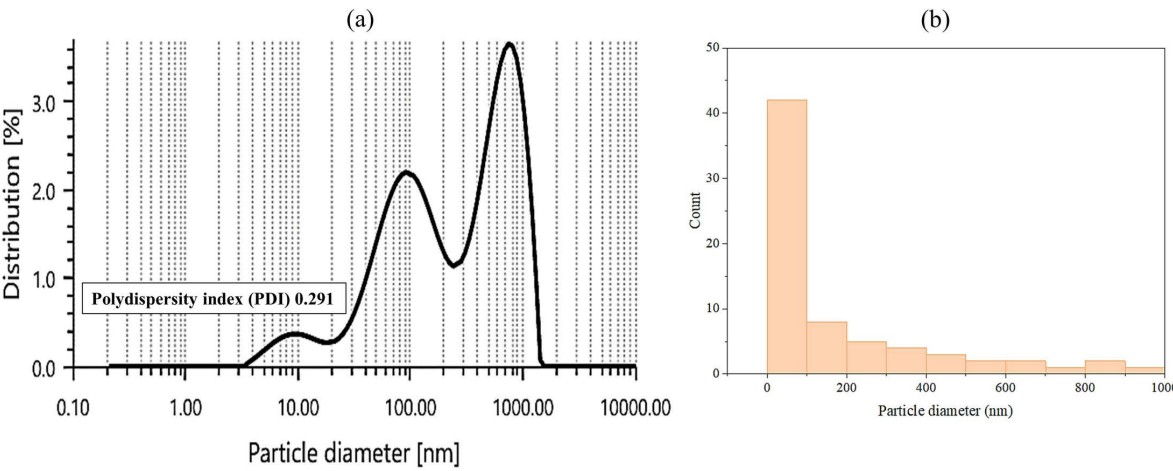

**Fig 3. Particle size determination of synthesized nanoparticles.** (a) size distribution by intensity (b) size distribution histogram based on number data.

Additionally, the outcomes demonstrated that the biosynthesized NPs exhibited a polydispersity index (PDI) of 0.291, which is adequate for applications involving drug transport [43]. Because PDI is a measure of a particle's stability and dispersity, monodispersed, stable nanoparticles were defined as having a PDI of less than 0.5 [44].

The surface charge and stability of the AgNPs were assessed using the zeta potential. The AgNPs that were synthesized showed a zeta potential of −38.6 mV (Fig 4). The behavior of nanoparticles in colloidal suspensions is governed



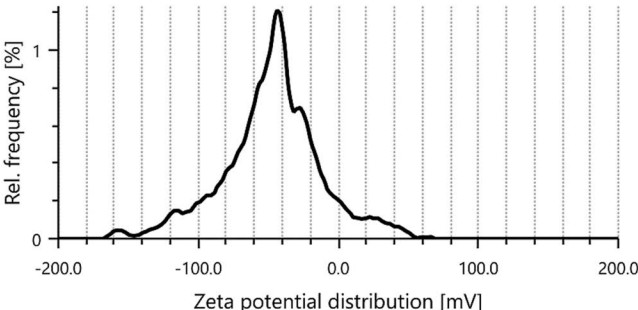

**Fig 4. Zeta potential measurement: Analysis of zeta potential of synthesized silver nanoconjugates of Linalool.**

by their zeta potential, which also shows variations in the surface and repulsive forces between the particles, particularly during the storage phase. Low extract content, alkaline pH, and increasing $AgNO_3$ concentration resulted in a decrease in the zeta potential. The surface charge, stability, and dispersion of AgNPs are indicated by their zeta potential values [45]. Because of interparticle repulsion, which prevents particles from aggregating when the zeta potential is greater than ± 30 mV, colloidal suspensions remain stable [46]. Owing to their significantly cationic and anionic surface charges, nanoparticles with zeta potentials greater than +30 mV or less than −30 mV indicate good stability, whereas those with zeta potentials between −10 mV and +10 mV are regarded as neutral [47]. In the current study, essential oil acted as a steric stabilizer on the AgNP surfaces, preventing the nanoparticles from clumping together. Consequently, we showed that the synthesis performed in these settings produces remarkably stable nanoparticles that do not clump or enlarge over time.

**3.1.3. Identification of chemical bonds using Fourier transform infrared spectroscopy (FT-IR).** As shown in Fig 5, the interaction between linalool and AgNPs in LN@AgNPs was examined by FTIR spectroscopy. The common absorption bands between linalool and LN@AgNPs were verified by FTIR analysis. The functional groups associated with these bands may play a role in stabilizing the AgNPs and reducing silver ions.

The FT-IR spectrum of LN@AgNPs exhibited a broad peak around 3300 cm$^{-1}$, corresponding to O–H stretching vibrations, indicating the presence of hydroxyl groups and possible interactions between AgNPs and linalool. Polar −OH groups are known to strongly coordinate with metal ions, facilitating the stabilization of silver nanoparticles [48]. A peak observed near 1600 cm$^{-1}$ was attributed to C=O stretching, suggesting interaction between AgNPs and the carbonyl groups of linalool. Additionally, a band around 1300 cm$^{-1}$ was associated with C–O vibrations, further confirming the molecular structure of linalool and its involvement in nanoparticle formation. These bands are commonly linked to metal chelation, which can contribute to reduced metal toxicity [49]. The FTIR data indicate that the phytochemicals are probably attached to the surface of the nanoparticles by means of carboxyl and hydroxyl functional groups. Proteins, phenols, terpenoids, and carboxylic acids are examples of plant-derived biomolecules that act as stabilizing and reducing agents, aiding in the production and surface functionalization of the nanoparticles [37]. These findings, which are consistent with earlier research, validate the synthesis of LN@AgNPs [50–52].

**3.1.4. Scanning electron microscopy (SEM).** The samples were subjected to SEM analysis to determine the particle shape and size distribution. This approach makes it possible to gather information on the size and shape of the nanoparticles, as well as quantitative and qualitative data [37]. These particles were small, solid, and somewhat elongated, with an average diameter ranging from 66 nm to 413 nm, according to LN@AgNP imaging. Randomly distributed nanoparticles were observed in the linalool-based nanoconjugates, as demonstrated by SEM images (Fig 6). The size and shape of AgNPs have been shown to have a major influence on their antibacterial activity in a number of investigations [3]. Scientists believe that the increased release of silver ions by these NPs is responsible for their strong antibacterial action [53].

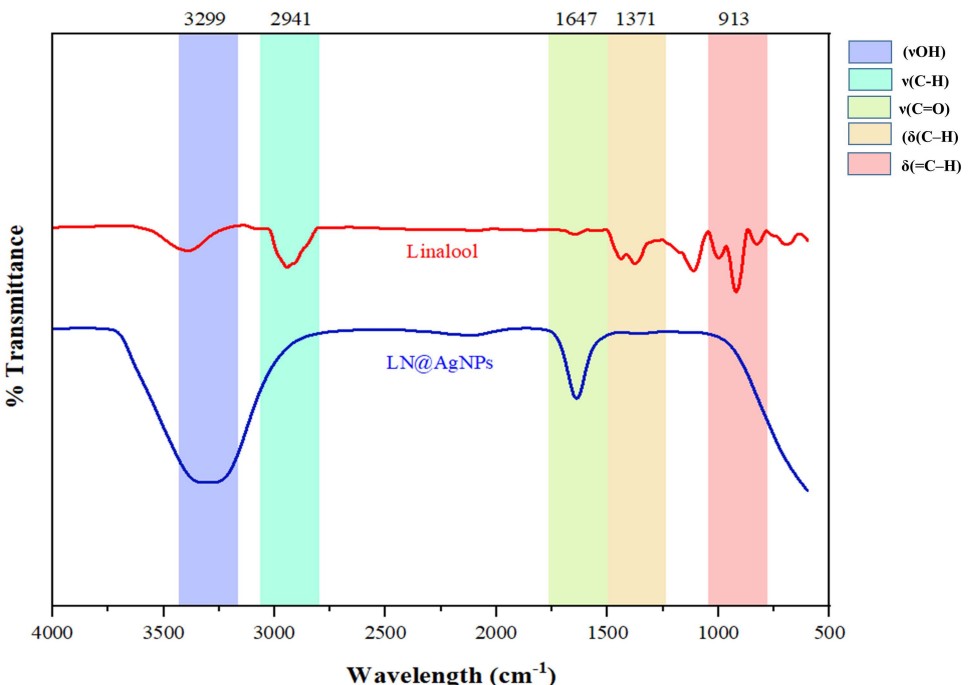

**Fig 5. FTIR spectrum of LN@AgNPs: Graph with wavelength (cm⁻¹) on the x-axis and transmittance percentage on the y-axis.**

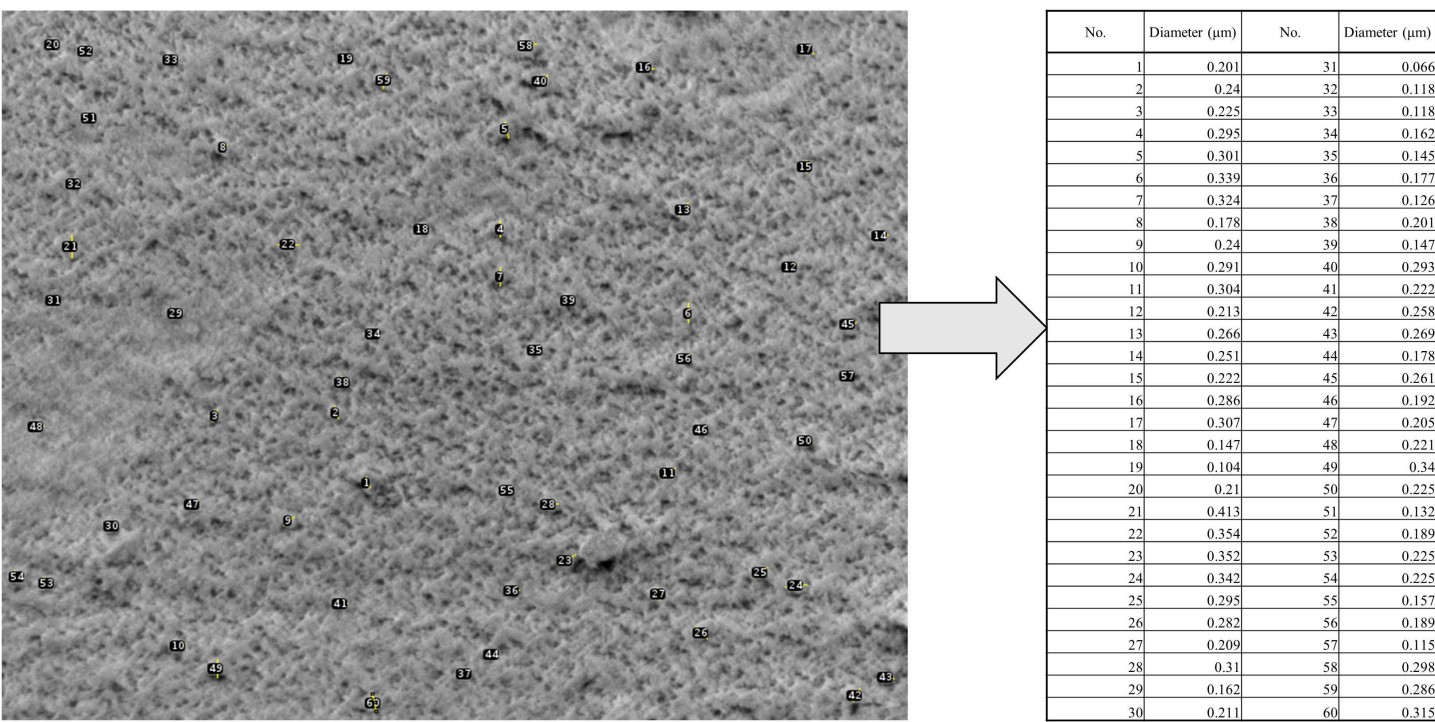

**Fig 6. SEM image of LN@AgNPs: Magnified image of selected area (left) with number of particles with respective area (right).**



The net intensity numbers obtained from Energy Dispersive X-ray (EDX) examination showed a proportional amount of X-ray counts for every element. Silver was more prevalent in the sample, as evidenced by the higher net intensity of silver (1763.52) compared to that of oxygen (251.45) and the higher weight percentage of silver. The longer oxygen peaks from the EDX of the nanoconjugate may further suggest the existence of the OH group, as they show that linalool was adsorbed on the nanoparticles (Fig 7).

## 3.2. Evaluation of antibacterial activity of Ag@LN NPs

### 3.2.1. *In silico* study (Molecular docking evaluation).

To enhance our understanding of the relationship between Linalool and LN@AgNPs with bacterial receptors, molecular docking was performed using the modeled 3D structures of Glyceraldehyde-3-phosphate dehydrogenase A (GAPDH) from *E. coli* (6IO4), FtsZ B. from *Bacillus subtilis* (2VAM), and type IVb pilin from *Salmonella enterica* (3FHU) (Fig 8, 9).

A strong beneficial connection between receptors and ligands in the most advantageous conformations is indicated by the lowest free energy value. We showed that by specifically targeting Cys, His, and Thr in the catalytic sites of target proteins, silver reduces their ability to operate as enzymes. Silver has long been employed as an antibacterial treatment; most of the proteins identified are linked to E. coli's core carbon metabolism, according to a previous study by Wang, H. *et al*. They investigated how silver targets Cys149 in GAPDH's catalytic region of GAPDH to reduce its enzymatic function. Ag+correlates with Cys149 and His176, according to the X-ray structure [54]. Among the 20 amino acids, cysteine, histidine, and threonine have the highest binding energies with silver ions [55].

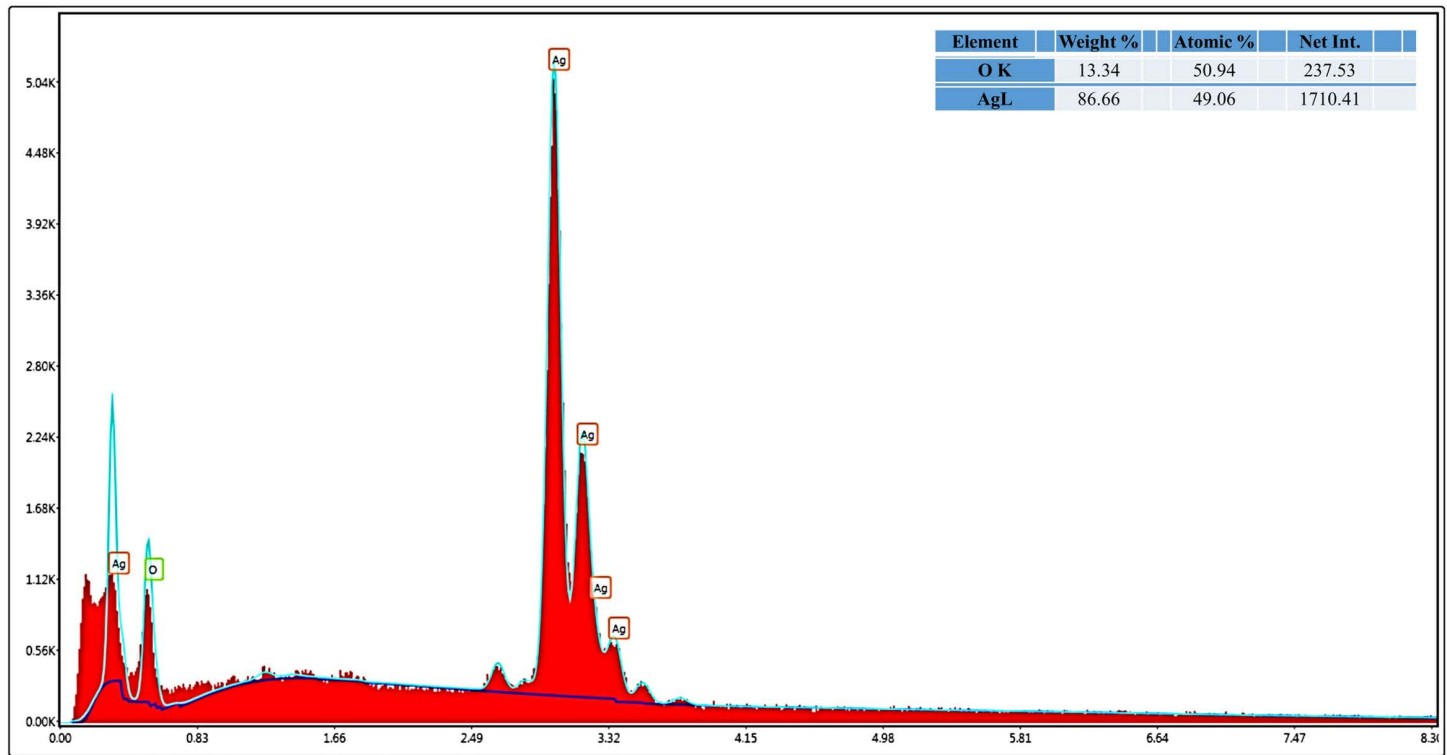

| Element | Weight % | Atomic % | Net Int. |
|---|---|---|---|
| O K | 13.34 | 50.94 | 237.53 |
| AgL | 86.66 | 49.06 | 1710.41 |

**Fig 7. EDAX spectrum of LN@AgNPs.**



**Fig 8. Interaction diagram of ligands.** (a-b LN@AgNPs, c. Linalool) to binding site of Glyceraldehyde-3-phosphate dehydrogenase A (GAPDH) of *E. coli* (6IO4).

LN@AgNPs showed favorable binding score of −4.5, −5.1 and −3.9 Kcal/mol for 6IO4, 2VAM and 3FHU respectively, while Linalool in individual exhibited −4.0, −4.5 and −3.3 Kcal/mol against 6IO4, 2VAM and 3FHU respectively which is less favourable then LN@AgNPs. The results are presented in Table 1. Ligand-protein visualization is shown in Fig 8, 9.

The best docking conformations for each of the LN@AgNPs and individual linalool were saved and graphically explored to examine the differences in docking scores caused by distinct interaction patterns. It was observed that both LN@AgNPs and Linalool were capable of forming hydrogen bonds with adjacent 6IO4 (*E.coli*) residues, but LN@AgNPs created a more stable complex by forming strong coordination bonds with the thiol group in cysteine, which is a strong electron donor (Fig 9a), and the imidazole group in histidine, which coordinated metal ions via its nitrogen atoms (Fig 9b). When we investigated ligands for the 2VAM protein of *Bacillus subtilis*, it was observed that individual linalool only formed one H-bond with Thr 133 residues while LN@AgNPs formed two H-bonds with the O-group to residues Asn166 and Thr133 in acceptor and donor motifs, respectively, and metal-acceptor bonds with residue Thr133, making LN@AgNPs more stable toward inhibition of the *Bacillus subtilis* FtsZ protein (Fig 10 a, b). The complex of 3FHU (*Salmonella enteric*) with LN@AgNPs was again more stable than the individual linalool (Fig 10c, d), as LN@AgNPs formed two H-bonds with the 3FHU protein residues, Thr115 and Thr117, in the acceptor



**Fig 9. Interaction diagram of ligands.** (a. LN@AgNPs, b. Linalool) to binding site of FtsZ B. of *Bacillus subtilis* (2VAM) and ligands (c. LN@AgNPs, d. Linalool) to binding site of type IVb pilin from *Salmonella enteric* (3FHU).

**Table 1. Molecular interaction Profile of LN@AgNPs and individual linalool with selected targets.**

| Strains | Receptor Protein | Ligand | Docking score (Kcal/mol) | Metal Acceptor | H-Bonds | H-Bonds Residues | Hydrophobic |
|---|---|---|---|---|---|---|---|
| *E.Coli* | 6IO4 | Ln@AgNPs | −4.5 | C149,H176 | 2 | C149,N313 | R10, P121,C149 |
| | | Linalool | −4.0 | – | 1 | E314 | R10,I11 |
| *Bacillus subtilis* | 2VAM | Ln@AgNPs | −5.1 | T133 | 2 | T133,N166 | F183 |
| | | Linalool | −4.5 | – | 1 | T133 | F183 |
| *Salmonella enteric* | 3FHU | Ln@AgNPs | −3.9 | T115,T117 | 2 | T115,T117 | V99,A101,V176 |
| | | Linalool | −3.3 | – | 1 | T115 | V99, A101 |

motif. The Ag atoms of LN@AgNPs formed two metal-acceptor bonds with Thr115 and Thr117. The findings of this study demonstrated that both linalool and LN@AgNPs inhibited the ability of the target proteins to function as enzymes by interacting with Cys, His, and Thr in their catalytic sites through hydrogen bonds. In the case of LN@AgNPs, this interaction (metallic chelation) was observed, increasing its potential as an antibacterial agent against

the proteins of particular strains. The understanding and creation of effective antibiotic therapies depend heavily on graphical analyses and structures [55].

**3.2.2. Establishing minimal inhibitory concentration.** The antibacterial activity was assessed using silver nanoconjugates. The minimum inhibitory concentration (MIC) and percentage inhibition of linalool and LN@AgNPs were determined using one gram-positive bacterial strain, *Bacillus subtilis*, and two gram-negative bacterial strains, *Salmonella enterica* and *Escherichia coli*, with ATCC numbers 6633, 14028 25922 respectively. The growth scores of the test wells were influenced by turbidity levels (Fig 10). Microtiter plate MIC assay was used against gram-negative bacterial strains *Salmonella enterica* and *Escherichia coli*, which were measured at 12.5 µL/mL, and against gram-positive bacterial strains *Bacillus subtilis*, linalool (MIC = 25 µL/mL), and LN@AgNPs (MIC = 6.25 µL/mL). The MIC values and the associated inhibition percentage, which were obtained by measuring the optical density (OD) at 560 nm, are shown in Table 2. These findings suggest that Gram-positive strains are more vulnerable than Gram-negative strains.

**3.2.3. Evaluation of agar well and disc diffusion susceptibility assays.** The antibacterial activity of the manufactured silver nanoconjugates based on the essential oil (linalool) is shown in Fig 11, 12. Azithromycin (AZM) was used as the positive control, while distilled water was used as the negative control. The investigation showed that

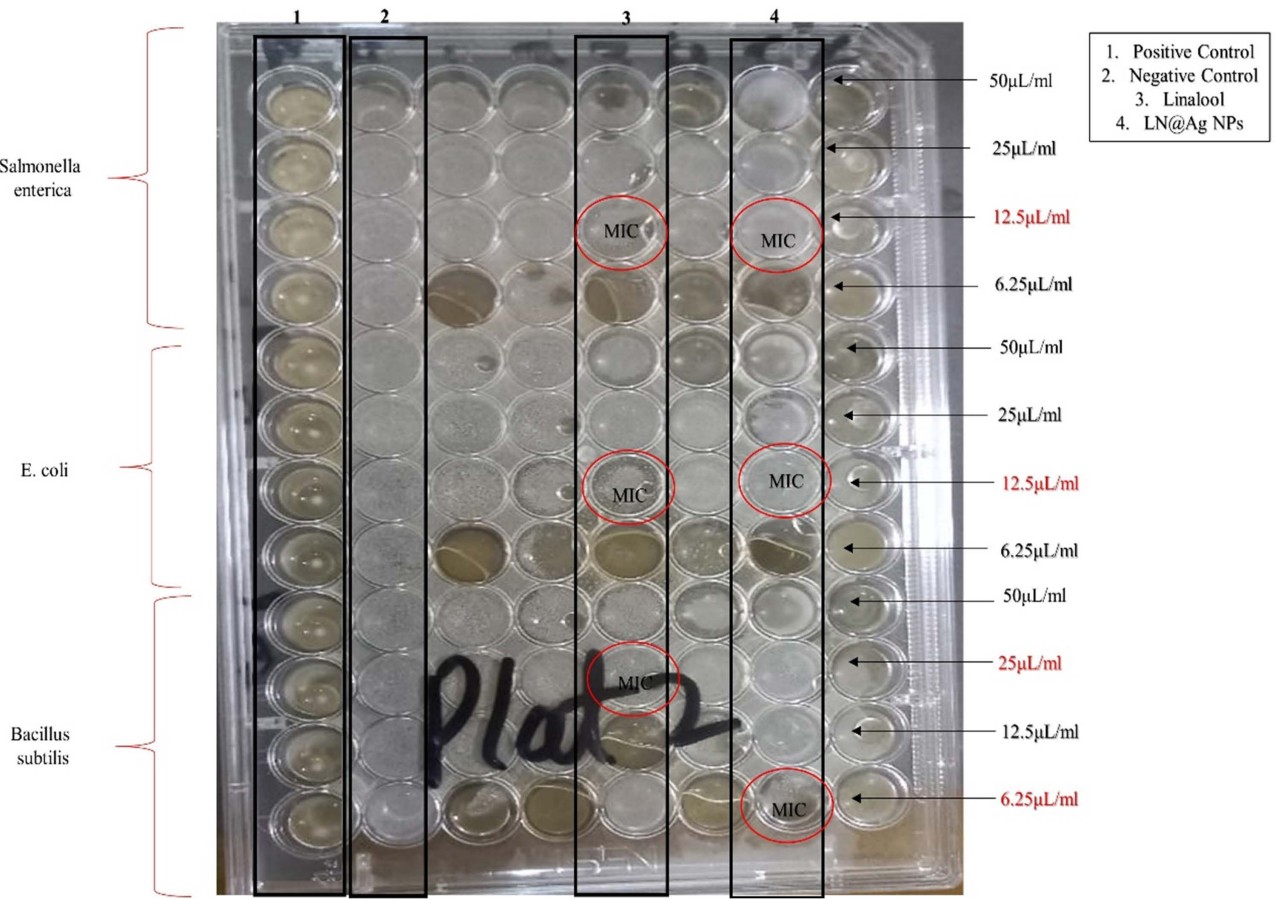

**Fig 10. A microtiter plate showing minimum inhibitory concentration (MIC): Score growth in test wells presence/absence of turbidity (1. Positive control, 2. Negative control, 3.Linalool and 4. LN@AgNPs) against Salmonella enterica, E.coli and Bacillus subtilis.**

**Table 2. Determination of MIC (µL/mL) and %Inhibition by microdilution method.**

| Strains | Linalool | | LN@AgNPs | |
|---|---|---|---|---|
| | MIC (µL/mL) | %I | MIC (µL/mL) | %I |
| *Salmonella enteric* | 12.5 | 73.17 | 12.5 | 98.82 |
| *Escherichia coli* | 12.5 | 91.8 | 12.5 | 86.96 |
| *Bacillus subtilis* | 25 | 99 | 6.25 | 86 |

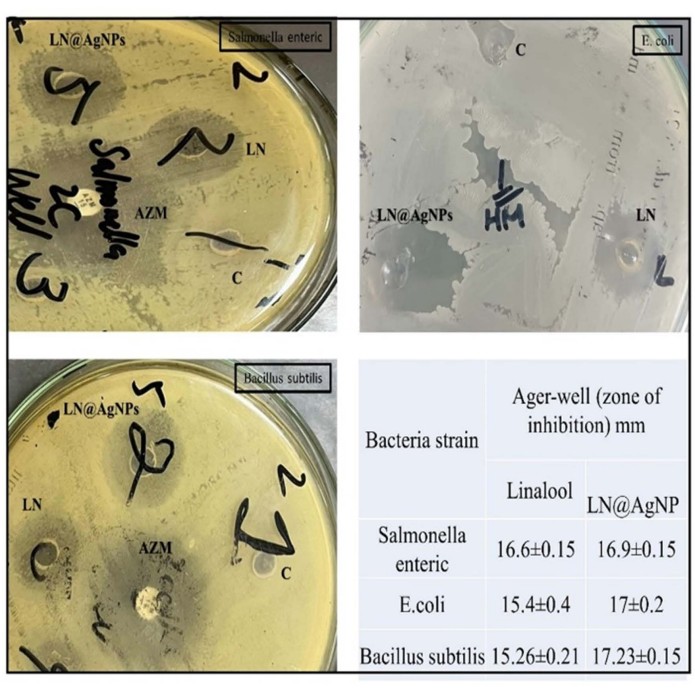

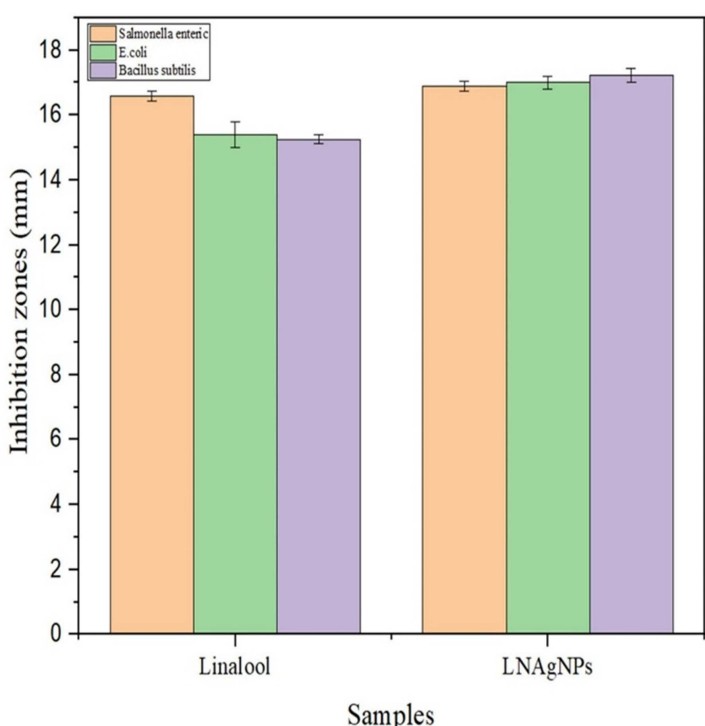

**Fig 11. Antibacterial activity evaluating using Agar-well diffusion method: micro plates for bacterial strains *Salmonella enterica, E.coli* and *Bacillus subtilis* with zone of inhibitions labeled with negative control (c), positive control (AZM), Linalool (LN) and LN@AgNPs (Left) and bar-graph showing zone of inhibitions (right).**

both linalool and LN@AgNPs had antibacterial activity; however, LN@AgNPs outperformed linalool alone in their ability to combat all bacterial types. Gram-positive (*Bacillus subtilis*) and gram-negative (*Salmonella enteric* and *Escherichia coli*) bacterial strains had restricted area diameters of 17.23, 16.9, and 17 mm. The free linalool exhibited restricted area diameters of 15.26, 16.6, and 15.4 mm respectively. This result implies that Gram-positive strains are more sensitive than Gram-negative strains when employing the agar well method (Fig 11). The LN@AgNP inhibitory zone in the disc diffusion assay had a diameter of 15–18 mm when tested against the strains under investigation (Fig 12). Linalool in its free form, as a small inhibitory zone, had a diameter of 10–14 mm against the study strains. Azithromycin (AZM), used as a positive control, produced inhibition zones of 17–18 mm across all strains in both assays, indicating that the antibacterial efficacy of LN@AgNPs was comparable to that of the standard antibiotic. Although standard AgNPs were not included for direct comparison, future studies should incorporate them to further validate the enhanced activity of LN@AgNPs.

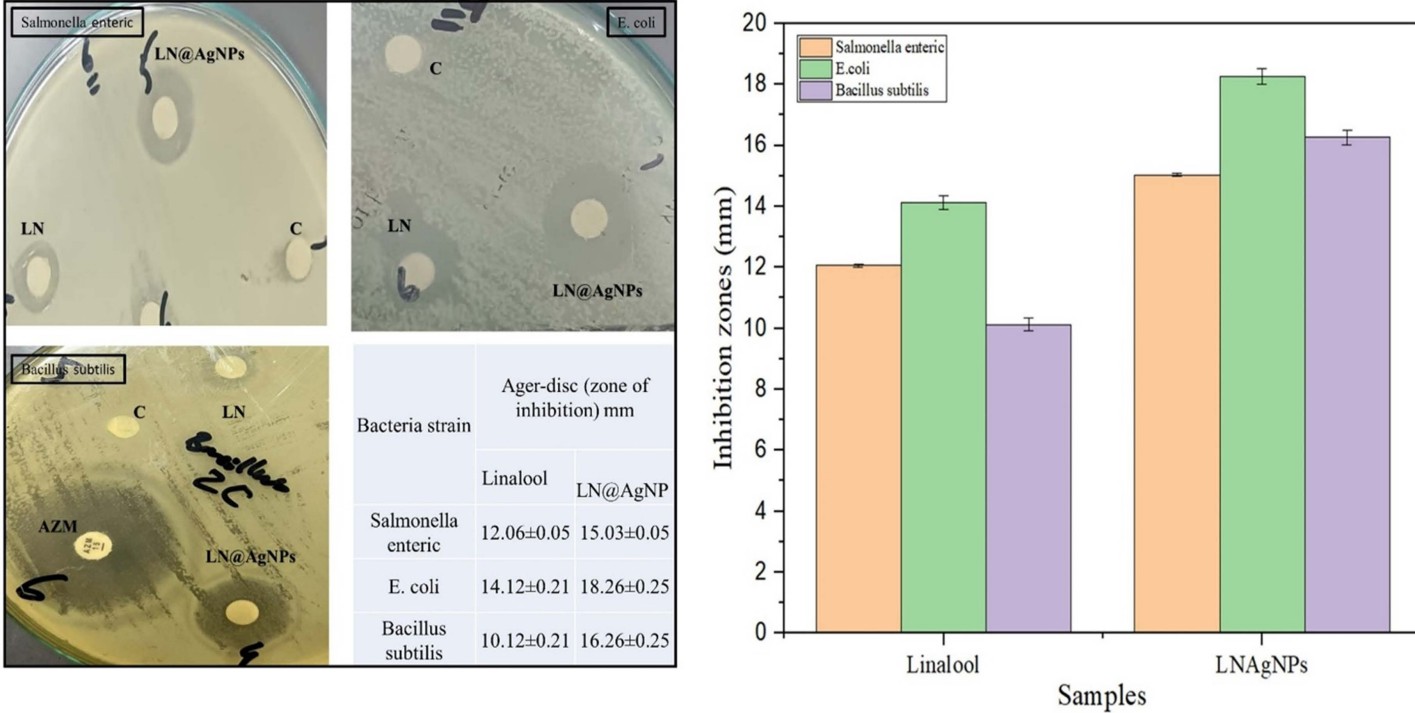

**Fig 12. Antibacterial activity evaluating using Disc diffusion method: micro plates for bacterial strains;** *Salmonella enterica, E.coli and Bacillus subtilis* **with zone of inhibitions labeled with negative control (c), positive control (AZM), Linalool (LN) and LN@AgNPs (Left) and bar-graph showing zone of inhibitions (right).**

These features can influence how effective they are as an antibacterial agent; in this work, we showed that, in comparison to individual component essential oil, synthesized LN@AgNPs with modest size had a strong effect against pathogenic microorganisms. In their study (Wang *et al*. 2019 [54]; Nie, D *et al* 2011 [56]; Durán *et al*. 2016 [57]), generally agree that when nanoparticles attach to a cell membrane, they interact with phosphorus and sulfur atoms, leading to bacterial lysis and metabolic failure. Similarly, the structure of the membrane is changed by nanoparticles, which changes its permeability and, as a result, the way that substances move through it, leading to bacterial lysis. The antibacterial activity of AgNPs can be explained by the generation of free radicals, which harm cell membranes [57]. In agreement with Cutro, A. C. [58] LN@AgNPs synthesized with EO (linalool) showed lower MIC and inhibition zones against *E. coli* and *Salmonella enterica* compared to *Bacillus subtilis*. It can be inferred from this result that EO-generated LN@AgNPs exhibited greater efficacy against gram-positive bacteria.

## 4. Conclusion

The effective synthesis and characterization of LN@AgNPs, as shown by UV–Vis spectroscopy, FT-IR, DLS, zeta potential, and SEM investigations, validated their advantageous physicochemical characteristics, such as their crystalline nature, well-controlled size, and surface stability. The application of CMC and linalool as stabilizing and capping agents also supports a biocompatible formulation approach. Strong connections between LN@AgNPs and bacterial protein targets were found by molecular docking studies, and *in vitro* antibacterial tests demonstrated increased effectiveness compared to free linalool. These results suggest that LN@AgNPs have the potential to be useful antibacterial agents in pharmaceutical applications. To promote clinical development, future research should focus on safety evaluation, mechanistic investigations, and *in vivo* validation.



## Author contributions

**Conceptualization:** Faiza Hassan, Muhammad Umer Khan.

**Data curation:** Hina Manzoor, Malik Arslan Ali.

**Formal analysis:** Hina Manzoor, Mohammad A. Alfhili, Shakeel Waqqar, Malik Arslan Ali.

**Investigation:** Shakeel Waqqar, Malik Arslan Ali, Faiza Hassan, Samiullah Khan.

**Methodology:** Hina Manzoor, Samiullah Khan.

**Project administration:** Samiullah Khan, Muhammad Umer Khan.

**Resources:** Shakeel Waqqar, Faiza Hassan.

**Software:** Shakeel Waqqar, Malik Arslan Ali, Samiullah Khan.

**Supervision:** Mohammad A. Alfhili, Muhammad Umer Khan.

**Validation:** Mohammad A. Alfhili, Malik Arslan Ali, Faiza Hassan, Samiullah Khan, Muhammad Umer Khan.

**Visualization:** Hina Manzoor.

**Writing – original draft:** Hina Manzoor, Faiza Hassan.

**Writing – review & editing:** Mohammad A. Alfhili, Shakeel Waqqar, Muhammad Umer Khan.

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
