## [Decision Letter · Decision Letter 0]

5 Jun 2025

PONE-D-25-12992Novel Linalool-Silver Nanoparticles: Synthesis, Characterization, and Dual Approach Evaluation via Computational Docking and Antibacterial AssaysPLOS ONE

Dear Dr. Khan,

Thank you for submitting your manuscript to PLOS ONE. After careful consideration, we feel that it has merit but does not fully meet PLOS ONE’s publication criteria as it currently stands. Therefore, we invite you to submit a revised version of the manuscript that addresses the points raised during the review process.

We look forward to receiving your revised manuscript.

Kind regards,

Pankaj Thakur

Academic Editor

PLOS ONE

Reviewers' comments:

Reviewer's Responses to Questions

**Comments to the Author**

1. Is the manuscript technically sound, and do the data support the conclusions?

Reviewer #1: Yes

Reviewer #2: Partly

2. Has the statistical analysis been performed appropriately and rigorously? 

Reviewer #1: I Don't Know

Reviewer #2: Yes

3. Have the authors made all data underlying the findings in their manuscript fully available?

Reviewer #1: Yes

Reviewer #2: Yes

4. Is the manuscript presented in an intelligible fashion and written in standard English?

Reviewer #1: Yes

Reviewer #2: No

5. Review Comments to the Author

Reviewer #1: The manuscript titled "Novel Linalool-Silver Nanoparticles: Synthesis, Characterization, and Dual Approach Evaluation via Computational Docking and Antibacterial Assays" by Khan et al. presents a potentially interesting study. However, before consideration for publication, the following major revisions are required:

1. The abstract is overly lengthy and should be made more concise. Please focus primarily on the key results and implications. Avoid extensive background or methodological detail in this section.

2. The introduction is too brief and lacks depth. It should be expanded to provide a more comprehensive background of the topic. Additionally, please incorporate recent references from 2024 and 2025 to reflect the current state of research and strengthen the rationale of the study.

3. The figures provided are of low resolution, and the embedded text is not legible. Please replace all figures with high-resolution versions (minimum 300 DPI) to ensure clarity and readability.

4. In my opinion, the Results and Discussion sections should be presented together rather than separately. Integrating these sections will allow for a more cohesive interpretation of the findings and help the reader better understand the significance of the results in context.

5. A detailed comparison of the present results with previously published studies is necessary and discusses how this work advances the field.

6. The manuscript’s language lacks fluency in certain sections. It requires substantial improvement in grammar, sentence structure, and scientific expression. A professional English editing service is recommended.

7. The conclusion section is too brief and lacks a comprehensive summary of the study’s key findings, implications, and future directions. The authors are advised to expand this section to clearly highlight the major results, significance of the work, and potential applications.

The study has potential but requires significant improvement in structure, language, presentation, and scientific depth. Addressing the above points will enhance the clarity, impact, and rigor of the manuscript.

Reviewer #2: This manuscript explores the synthesis and characterization of linalool-functionalized silver nanoparticles (LN@AgNPs) and evaluates their antimicrobial activity via both in vitro assays and in silico molecular docking studies. The combination of natural essential oil-based nanoconjugates with silver nanoparticles is an important area of research, particularly in the context of increasing antibiotic resistance. The study is methodologically sound, and the manuscript is generally well-structured. However, a number of issues related to clarity, data interpretation, and scientific depth need to be addressed before this manuscript can be considered for publication.

1. The particle size distribution is highly polydisperse (ranging from ~9 to 680 nm). This raises concerns about batch uniformity, which affects biological reproducibility.

2.Regarding FT IR spectra-Include complete spectra and assign peaks precisely.

3. Regarding Antibacterial Assays-Positive controls (AZM) are included, but no comparison with standard AgNPs was provided.

4. Regarding Grammar & Language: The manuscript contains several grammatical and typographical errors. Some examples:

“...make it more potential...” → should be “...makes it more potent...”

“...examined for antibacterial in-silico and in-vitro studies...” → revise for clarity.

Kindly provide the appropriate answers to the above suggestions

6. PLOS authors have the option to publish the peer review history of their article (what does this mean? ). If published, this will include your full peer review and any attached files.

**Do you want your identity to be public for this peer review?** For information about this choice, including consent withdrawal, please see our Privacy Policy .

Reviewer #1: **Yes: ** ANKUSH GUPTA

Reviewer #2: No

---

## [Author Response · Author response to Decision Letter 1]

17 Jun 2025

We sincerely thank the editor and reviewers for their detailed and constructive feedback on our manuscript. We have carefully addressed all comments and revised the manuscript accordingly. Below, we provide a point-by-point response to each comment.

Reviewer 1 Comments:

Comment 1: The abstract is overly lengthy and should be made more concise. Please focus primarily on the key results and implications. Avoid extensive background or methodological detail in this section.

Response: We acknowledge the reviewer’s comment. The abstract has been revised to focus more on the key findings and their implications while minimizing background and methodological details.

Comment 2: The introduction is too brief and lacks depth. It should be expanded to provide a more comprehensive background of the topic. Additionally, please incorporate recent references from 2024 and 2025 to reflect the current state of research and strengthen the rationale of the study.

Response: Thank you for your valuable comment. We have expanded the introduction to provide a more comprehensive background by highlighting the role of nanoparticles in pathogen control, the antibacterial potential of AgNPs, and the activity of linalool-rich essential oils. Recent literature from 2024 and 2025 has also been incorporated to reflect current research.

Comment 3: The figures provided are of low resolution, and the embedded text is not legible. Please replace all figures with high-resolution versions (minimum 300 DPI) to ensure clarity and readability.

Response: We sincerely appreciate the reviewer’s feedback. In response, we have significantly enhanced the quality of all figures using the PACE image optimization tool. Each figure has now been uploaded in high resolution (600 dpi) and in TIFF format, as per standard publication requirements.

Comment 4: In my opinion, the Results and Discussion sections should be presented together rather than separately. Integrating these sections will allow for a more cohesive interpretation of the findings and help the reader better understand the significance of the results in context.

Response: Thank you for the constructive suggestion. The Results and Discussion sections have been merged into a unified section. This integrated format allows for a more fluid interpretation of the findings and aligns better with the manuscript’s narrative flow.

Comment 5: A detailed comparison of the present results with previously published studies is necessary and discusses how this work advances the field.

Response: We acknowledge the reviewer’s comment. We have now included a comparative discussion in the revised manuscript, highlighting how our findings align with previously published studies. This comparison also emphasizes the novelty and advancement of our work in the context of current antibacterial nanoparticle research.

Comment 6: The manuscript’s language lacks fluency in certain sections. It requires substantial improvement in grammar, sentence structure, and scientific expression. A professional English editing service is recommended.

Response: We sincerely appreciate the reviewer’s feedback. In response, the manuscript's English language has been thoroughly improved using the Paperpal editing tool to ensure clarity and readability.

Comment 7: The conclusion section is too brief and lacks a comprehensive summary of the study’s key findings, implications, and future directions. The authors are advised to expand this section to clearly highlight the major results, significance of the work, and potential applications.

Response: We are grateful for the thoughtful insights. We have expanded the conclusion to provide a more comprehensive summary of the key findings, emphasize the significance of the work, and outline its potential applications and future research directions.

Reviewer 2 Comments:

Comment 1: The particle size distribution is highly polydisperse (ranging from ~9 to 680 nm). This raises concerns about batch uniformity, which affects biological reproducibility.

Response: We apologize for the lack of clarity in the manuscript. The wording did not accurately reflect our analysis. We have now clarified that Figure 3a presents size distribution data based on intensity from DLS analysis. Although the DLS intensity distribution showed a dominant peak at ~630 nm, this does not necessarily indicate that most particles were of that size. Due to the R⁶ dependence of light scattering, even a small population of large particles can significantly influence intensity measurements. To address this, we constructed a histogram (Figure 3b) using number-based data derived from the same dataset. This revealed that the highest frequency of particles falls below 100 nm, with an average size of approximately 89 nm calculated using Gaussian fitting and nonlinear curve analysis. Furthermore, the PDI value of 0.29 supports the moderate monodispersity and batch uniformity of the sample, indicating suitability for biological applications.

Comment 2: Regarding FT IR spectra-Include complete spectra and assign peaks precisely.

Response: Thank you for the observation. We have now included the complete FT-IR spectra in the revised manuscript and have precisely assigned the characteristic peaks. Additionally, we have added the FT-IR spectrum of pure linalool alongside that of LN@AgNPs to enable direct comparison and better highlight the functional groups involved in nanoparticle formation. We have revised the description with precise peak assignments, correlating each observed FT-IR band with specific functional groups present in linalool and their interaction with silver nanoparticles.

Comment 3: Regarding Antibacterial Assays-Positive controls (AZM) are included, but no comparison with standard AgNPs was provided.

Response: Thank you for your valuable comment. We have revised the manuscript to address this point and now provide a comparison of LN@AgNPs with azithromycin (AZM) as a standard antibacterial reference.

Comment 4: Regarding Grammar & Language: The manuscript contains several grammatical and typographical errors. Some examples:

“...make it more potential...” → should be “...makes it more potent...”

“...examined for antibacterial in-silico and in-vitro studies...” → revise for clarity.

Response: We sincerely appreciate the reviewer’s feedback. In response, the manuscript's English language has been thoroughly improved using the Paperpal editing tool to ensure clarity and readability.

---

## [Decision Letter · Decision Letter 1]

17 Sep 2025

Novel Linalool-Silver Nanoparticles: Synthesis, Characterization, and Dual Approach Evaluation via Computational Docking and Antibacterial Assays

PONE-D-25-12992R1

Dear Dr. Khan,

We’re pleased to inform you that your manuscript has been judged scientifically suitable for publication and will be formally accepted for publication once it meets all outstanding technical requirements.

Kind regards,

Pankaj Thakur

Academic Editor

PLOS ONE

Additional Editor Comments (optional):

Reviewers' comments:

Reviewer's Responses to Questions

**Comments to the Author**

1. If the authors have adequately addressed your comments raised in a previous round of review and you feel that this manuscript is now acceptable for publication, you may indicate that here to bypass the “Comments to the Author” section, enter your conflict of interest statement in the “Confidential to Editor” section, and submit your "Accept" recommendation.

Reviewer #3: All comments have been addressed

2. Is the manuscript technically sound, and do the data support the conclusions?

Reviewer #3: Yes

3. Has the statistical analysis been performed appropriately and rigorously? 

Reviewer #3: Yes

4. Have the authors made all data underlying the findings in their manuscript fully available?

Reviewer #3: Yes

5. Is the manuscript presented in an intelligible fashion and written in standard English?

Reviewer #3: Yes

6. Review Comments to the Author

Reviewer #3: The manuscript titled “Novel Linalool-Silver Nanoparticles: Synthesis, Characterization, and Dual Approach Evaluation via Computational Docking and Antibacterial Assays” presents the synthesis of linalool-conjugated silver nanoparticles and evaluates them using both computational docking and antibacterial assays. The study is well-designed and methodologically sound, addressing a significant area of nanobiotechnology with important biomedical implications. By integrating in silico docking studies with in vitro antibacterial assays, the authors provide a comprehensive evaluation that enhances the impact of their work. The nanoparticles are thoroughly characterized through UV-Vis, FTIR, SEM, XRD, DLS, and zeta potential analyses, effectively demonstrating their stability and uniformity. Antibacterial activity is assessed systematically against both Gram-positive and Gram-negative bacterial strains, with clear evidence that LN@AgNPs exhibit superior performance compared to free linalool. The discussion is well-structured, successfully integrating findings with relevant literature and underscoring the novelty of linalool-conjugated AgNPs as antimicrobial agents. The manuscript has also been meticulously revised in response to earlier reviewer feedback, resulting in improvements in figure quality, language fluency, and organization. Importantly, there are no ethical concerns associated with this work, and the “N/A” declaration for animal and human studies is appropriate. The authors clearly state that no funding was received and declare no competing interests. The manuscript appears original, with no evidence of dual publication or overlap with prior work. Overall, this is a well-executed and clearly presented study that contributes valuable insights to the field of nanomaterial-based antibacterial therapies. The findings convincingly demonstrate the potential of linalool-silver nanoconjugates as effective antimicrobial agents, supported by robust evidence from both computational and experimental approaches. I recommend the acceptance of this manuscript in its present form.

7. PLOS authors have the option to publish the peer review history of their article (what does this mean? ). If published, this will include your full peer review and any attached files.

**Do you want your identity to be public for this peer review?** For information about this choice, including consent withdrawal, please see our Privacy Policy .

Reviewer #3: No

---

## [Editor Report · Acceptance letter]

PONE-D-25-12992R1

PLOS ONE

Dear Dr. Khan,

I'm pleased to inform you that your manuscript has been deemed suitable for publication in PLOS ONE. Congratulations! Your manuscript is now being handed over to our production team.

Kind regards,

on behalf of

Prof Pankaj Thakur

Academic Editor

PLOS ONE